# Transcriptome and Metabolome Analyses Reveal That Jasmonic Acids May Facilitate the Infection of Cucumber Green Mottle Mosaic Virus in Bottle Gourd

**DOI:** 10.3390/ijms242316566

**Published:** 2023-11-21

**Authors:** Zhenggang Li, Yafei Tang, Guobing Lan, Lin Yu, Shanwen Ding, Xiaoman She, Zifu He

**Affiliations:** Guangdong Provincial Key Laboratory of High Technology for Plant Protection, Plant Protection Research Institute, Guangdong Academy of Agricultural Sciences, Guangzhou 510640, China; lizhenggang@gdppri.com (Z.L.); yf.tang1314@163.com (Y.T.); languo020@163.com (G.L.); yulin@gdppri.com (L.Y.); dingshanwen@gdppri.com (S.D.)

**Keywords:** cucumber green mottle mosaic virus, jasmonic acids, transcriptome and metabolome, infection, bottle gourd

## Abstract

Cucumber green mottle mosaic virus (CGMMV) is a typical seed-borne tobamovirus that mainly infects cucurbit crops. Due to the rapid growth of international trade, CGMMV has spread worldwide and become a significant threat to cucurbit industry. Despite various studies focusing on the interaction between CGMMV and host plants, the molecular mechanism of CGMMV infection is still unclear. In this study, we utilized transcriptome and metabolome analyses to investigate the antiviral response of bottle gourd (*Lagenaria siceraria*) under CGMMV stress. The transcriptome analysis revealed that in comparison to mock-inoculated bottle gourd, 1929 differently expressed genes (DEGs) were identified in CGMMV-inoculated bottle gourd. Among them, 1397 genes were upregulated while 532 genes were downregulated. KEGG pathway enrichment indicated that the DEGs were mainly involved in pathways including the metabolic pathway, the biosynthesis of secondary metabolites, plant hormone signal transduction, plant–pathogen interaction, and starch and sucrose metabolism. The metabolome result showed that there were 76 differentially accumulated metabolites (DAMs), of which 69 metabolites were up-accumulated, and 7 metabolites were down-accumulated. These DAMs were clustered into several pathways, including biosynthesis of secondary metabolites, tyrosine metabolism, flavonoid biosynthesis, carbon metabolism, and plant hormone signal transduction. Combining the transcriptome and metabolome results, the genes and metabolites involved in the jasmonic acid and its derivatives (JAs) synthesis pathway were significantly induced upon CGMMV infection. The silencing of the *allene oxide synthase* (*AOS*) gene, which is the key gene involved in JAs synthesis, reduced CGMMV accumulation. These findings suggest that JAs may facilitate CGMMV infection in bottle gourd.

## 1. Introduction

Cucumber green mottle mosaic virus (CGMMV) is a typical seed-borne plant virus belonging to the genus *Tobamovirus*. CGMMV is known to cause significant damage to cucurbit crops such as cucumber, watermelon, bottle gourd, squash, pumpkin, melons, and various gourd species [1,2,3,4,5,6]. Although the characteristic symptoms of CGMMV were first recorded in 1923 [7], it was not until 1935 that it was officially described in cucumber [8]. Since its initial discovery in England [8], CGMMV has rapidly spread globally due to the development of international trade [3,9,10,11,12,13,14,15,16,17]. In China, CGMMV was first detected in pumpkin in Guangxi province in 2005 [18], and has since caused widespread devastation to cucurbit crops across the country [19,20,21]. The marketable yield losses caused by CGMMV can be up to 50% and even 100% due to poor quality [12,21]. In 2006, CGMMV was listed as a national agricultural plant quarantine pest in China. 

Phytohormones play a crucial role in plant development and response to abiotic and biotic stresses. Jasmonic acid (JA) and its derivatives (JAs) are important phytohormones that regulate various physiological processes, including plant growth, metabolism, and stress responses against pathogens and herbivores [22,23,24]. The active form of JA hormone, jasmonoyl-isoleucine (JA-Ile), is synthesized in the chloroplast from α-linoleic acid or hexadecatrienoic acid (16:3) and requires three reaction sites: chloroplast, peroxisome, and cytoplasm [25,26,27]. Various enzymes are involved in the chemical reactions of JAs, including allene oxide synthase (AOS), allene oxide cyclase (AOC), oxo-phytodienoic acid reductase (OPR3), jasmonate resistant 1 (JAR1), JA carboxyl methyltransferase (JMT), and JASSY protein [26,28]. Upon stimulation by biotic or abiotic factors, JA-Ile is quickly synthesized in the cytoplasm and transported to the nucleus by jasmonic transfer protein 1 (JAT1). In the nucleus, JA-Ile binds CORONATINE INSENSITIVE1 (COI1) and jasmonate ZIM-domain protein (JAZ), leading to ubiquitination and degradation by the 26S proteasome pathway. The degradation of JAZ releases the binding sites, which makes the transcription factors (TFs) activate the expression of JA-responsive genes [24]. 

JAs have also been reported to be involved in response to virus infection, but their function varies in response to different viruses [29,30,31,32,33]. The rice stripe virus coat protein (CP) induces the JAs signaling pathway, which further upregulates the expression of MYB TFs to active ARGONAUTE 18 (AGO18)-mediated RNA silencing and antiviral defense in rice [31,34]. The C2 protein encoded by tomato yellow leaf curl Sardinia virus (TYLCSV) affects the activity of the Skp1/Cullin/F-box (SCF) complex by interacting with the CSN5 protein, which further inhibits JAs signaling in *Arabidopsis thaliana* [35]. However, exogenously applied JA reduces the local resistance of *N* gene-containing tobacco to tobacco mosaic virus (TMV) and silencing of COI1, or AOS reduces TMV accumulation, demonstrating that JA negatively regulates the resistance to tobacco mosaic virus in tobacco [36]. But, the function of the JAs signaling pathway under CGMMV stress is still unclear. 

CGMMV mainly infects cucurbit leaves, fruits, and seeds, causing mottling and mosaic symptoms on the leaves and fruit peel, brown necrotic lesions on stems and peduncles, and yellowing and spongy fruit flesh [10]. Previous studies using high-throughput deep sequencing have demonstrated that the infection of CGMMV in watermelon affects the expression of miRNAs and genes involved in cell wall modulation, the plant hormone signaling pathway, photosynthesis, primary and secondary metabolism, and intracellular transport [37,38]. An analysis of CGMMV-derived siRNAs indicated that different cucurbit species respond differently to CGMMV infection [39,40,41]. Transcriptome analysis of watermelon leaves and fruit under CGMMV stress revealed that DEGs are involved in photosynthesis, plant–pathogen interactions, secondary metabolism, and plant hormone signal transduction [38,42]. However, there has not been a study on the interaction between bottle gourd and CGMMV. 

Previously, we found that CGMMV has been one of the main viruses threatening bottle gourd in Guangdong province, China [43]. The infectious cDNA clone of CGMMV was constructed and the infectivity in different cucurbit crops was analyzed [44]. In this study, we used the transcriptome and metabolome to analyze the antiviral response of bottle gourd with CGMMV infection. Then, we focused on the hormone pathway, especially the JAs signaling pathway.

## 2. Results 

### 2.1. Inoculation and Virus Detection of CGMMV in Bottle Gourd

To inoculate bottle gourd plants, the cotyledons were infiltrated with *Agrobacterium* containing CGMMV infectious cDNA clones [44], while the mock plants were infiltrated with *Agrobacterium* containing an empty vector. At 12 days post inoculation (dpi), CGMMV infected plants developed obvious mosaic and mottling symptoms on the upper leaves compared to the mock plants (Figure 1A). The leaves of both the CGMMV-infected and mock plants were taken and subjected to RT-PCR and Western blot detection. RT-PCR using the CGMMV coat protein (CP) primer pair (Appendix A) showed the expected band, and Western blot with CGMMV CP antibody also revealed a specific band of the expected size (Figure 1B). 

### 2.2. Quality Control of RNA-Seq Data

To perform transcriptome and metabolome tests, the upper leaves of CGMMV-infected (ZLV12) and mock plants (ZLM12) were taken at 12 dpi, and three replicates were conducted in each treatment. The total RNA of all samples was extracted and subjected to deep sequencing using the Illumina HiSeq platform. The deep sequencing generated 298,190,450 raw reads and 291,823,170 (97.86%) clean reads after the removal of low-quality reads, contamination, and adapter sequences (Table 1). The clean reads encompassed about 43.76 Gb clean data, which were sufficient for gene expression analysis. Then, the clean data were mapped to the reference genome of bottle gourd (http://cucurbitgenomics.org/ftp/genome/BottleGourd/USVL1VR-Ls/) (accessed on 5 June 2022) using HISAT2 software (v2.1.0). The results showed that the unique mapped percentage of each replicate was over 92%, the Q20 percentage was over 97%, and the Q30 percentage was over 92%. The GC content of each replicate was over 44% (Table 1). 

### 2.3. Gene Expression Analysis of Bottle Gourd in Response to CGMMV Infection 

To identify the genes involved in response to CGMMV infection in bottle gourd, DEGs were analyzed using RNA-seq data. Different gene expression analyses between the mock and CGMMV treatments were performed using DESeq2 software (v1.22.2), which requires data of unnormalized reads. The genes with a false discovery rate (FDR) < 0.05 and |log_2_Fold Change| ≥ 1 were filtered as significant DEGs. A total of 1929 DEGs were identified between the mock and CGMMV treatments: 1397 DEGs were upregulated and 532 were downregulated (Figure 2A). The hierarchical clustering of DEGs in Figure 2B shows an overview of the transcriptome result. 

To cluster these DEGs, Gene Ontology (GO) pathway enrichment was performed. In the biological process of GO enrichment, the pathways related to response to drugs, regulation of hormone levels, response to ethylene, hormone metabolic process, innate immune response, and response to oxidative stress were the most enriched (Appendix A). The intrinsic component of plasma membrane was the only enriched pathway in the cellular component. In terms of the molecular function of GO, the significantly enriched pathways included tetrapyrrole binding and heme binding (Appendix A). 

Based on the GO analysis, the Kyoto Encyclopedia of Genes and Genomes (KEGG, https://www.genome.jp/kegg) (accessed on 5 June 2022) was used to further cluster the DEG pathways. Among the KEGG enrichment pathways, the metabolic pathway, the biosynthesis of secondary metabolites, plant hormone signal transduction, plant–pathogen interaction, and starch and sucrose metabolism were the most enriched (Figure 2C). 

### 2.4. Real-Time RT-PCR Verification of Transcriptome Result

To verify the RNA-seq analysis, 20 DEGs were selected to analyze the expressions after virus infection. The thirteen upregulated DEGs were Lsi01G013470, Lsi02G017750, Lsi05G011760, Lsi04G015060, Lsi02G007470, Lsi02G018990, Lsi01G009350, Lsi02G021080, Lsi02G008430, Lsi11G012260, Lsi01G014560, Lsi11G005040, and Lsi05G018940. The seven downregulated DEGs were Lsi05G012660, Lsi04G002080, Lsi03G008210, Lsi09G006560, Lsi02G013460, Lsi11G000250, and Lsi03G014380. The annotation of these DEGs (Table 2) was acquired by searching the bottle gourd genome website (http://www.cucurbitgenomics.org/search/genome/13) (accessed on 5 June 2022) using these accession numbers. The qRT-PCR results showed that all of these selected DEGs exhibited a similar trend to the transcriptome analysis, suggesting that the transcriptome analysis was convincing (Figure 3).

### 2.5. Metabolite Accumulation Analysis of Bottle Gourd in Response to CGMMV Infection 

To investigate the changes in metabolite accumulation under CGMMV stress, metabolome analysis was conducted between the mock and CGMMV-treated samples. The metabolites with |log_2_Fold Change| ≥ 1 were filtered as significant DAMs. A total of 76 DAMs were identified between the mock and CGMMV groups: 69 were significantly upregulated and 7 were significantly downregulated (Figure 4A). The most upregulated DAMs were 4-caffeoylquinic acid, acteoside, and malvidin-3,5-O-diglucoside (Appendix A). The most downregulated DAMs were glucarate O-phosphoric acid, cyclic AMP, and scopoletin (7-Hydroxy-5-methoxycoumarin) (Appendix A). The clustering heatmap of DAMs is shown in Figure 4B. To cluster these DAMs, KEGG pathway enrichment was performed based on the function of each DAM. KEGG showed that the biosynthesis of secondary metabolites, tyrosine metabolism, flavonoid biosynthesis, and the plant hormone signal transduction pathways were the most enriched (Figure 4C). 

### 2.6. Joint Analysis of Transcriptome and Metabolome of Bottle Gourd in Response to CGMMV Infection 

Based on the transcriptome and metabolome results, combined analysis was performed via KEGG enrichment. KEGG enrichment showed that plant hormone signal transduction, isoflavonoid biosynthesis, the biosynthesis of secondary metabolites, and alpha-linolenic acid metabolism were the most enriched pathways (Appendix A). 

Among these enriched pathways, the plant hormone signal transduction pathway was significantly enriched, as seen in transcriptome and metabolome analysis. The plant hormones consist of auxin, cytokinine, gibberellin, abscisic acid, ethylene, brassinosteroid, jasmonic acid, and salicylic acid. In the auxin signaling pathway, *auxin response factor 5* (ARF5) (Lsi04G017260) was upregulated about two times more in CGMMV-infected plants compared with the mock plants. However, the auxin hormone showed no significant change upon CGMMV infection. In the brassinosteroid biosynthesis pathway, the expression of *brassinosteroid insensitive 1* (BRI1) (Lsi11G000250) decreased by nearly 70% with CGMMV infection, while *BRI1 kinase inhibitor 1* (BKI1) (Lsi08G014580), *brassinosteroid insensitive 2* (BIN2) (Lsi01G002340), and brassinosteroid resistant 1/2 (BZR1/2) (Lsi07G002850) were induced about three times, two times, and two times, respectively (Figure 5). In addition, the expression of NPR1 and PR1 in the salicylic acid pathway showed no obvious response to CGMMV infection (Figure 5). 

In summary, joint analysis of DEGs and DAMs showed that the jasmonic acid and brassinosteroid signal transduction pathways were remarkably induced, while changes in other hormone signaling pathways were not obvious. 

### 2.7. Analysis of JAs signaling Pathway in Response to CGMMV Infection 

Among the plant hormone signal transduction pathways, the stimulation of the JAs synthesis pathway is the most prominent. The genes involved in the JAs synthesis pathways include *lipoxygenase* (*LOX*), *AOS*, *AOC*, *OPR3*, *JAR1*, *JMT*, *CYP94B3*, and *CYP94C1* (Figure 6A). qRT-PCR analysis showed that all these genes were upregulated upon CGMMV infection (Figure 6B). Moreover, the accumulation of JA and JA-Ile hormones was also remarkably increased (Figure 6C), demonstrating that the JAs signaling pathway was induced upon CGMMV infection.

In the JAs signaling pathway, JA-Ile will be transported to the nucleus from the cytoplasm to release the TFs to activate the expression of defensive genes. To identify the expression of TFs related with the JAs signaling pathway under CGMMV stress, the transcriptome data of TFs (Appendix A) were screened and analyzed. To validate the RNA-seq results, we selected various types of TFs that may be involved in the JAs signaling pathway and confirmed their expression levels using qRT-PCR. As expected, the transcription factor *Myc2* (Lsi09G016530), which can interact with JAZ protein, was induced about two times upon CGMMV infection, while another JAZ interacting transcription factor, *Myc3* (Lsi07G011090), was upregulated more than 30 times (Figure 7A). TFs *Myb13* (Lsi05G010160), *Myb62* (Lsi02G001830), *Myb77* (Lsi04G015930), and *Myb86* (Lsi06G015850) were also induced via CGMMV stimulation, but the expression of *Myb48* (Lsi10G013470) was reduced (Figure 7B). The TF *NAC* (Lsi08G004490) was also induced about nine times after CGMMV infection (Figure 7C). WRKY TFs play an important role in plant development, senescence, and response to abiotic and biotic stimuli. Compared with the mock control, the expression of *WRKY18* (Lsi02G012150) was slightly higher, *WRKY31* (Lsi05G014110) increased twice, and *WRKY68* (Lsi06G014830) increased nearly tenfold, but *WRKY6* (Lsi05G021100) decreased about twice (Figure 7D). ERF TFs are also modulated by the JAs signaling pathway. *ERF-RAP2* (Lsi08G002460) increased nearly one thousand times, *ERF1b* (Lsi09G012390) increased more than 30 times, *ERF1* (Lsi06G008160) and *ERF2* (Lsi07G001170) increased about 6–7 times, and *ERF2b* (Lsi03G010920) increased about two times, but *ERF6* (Lsi04G020490) decreased to about one third (Figure 7E). 

Taken together, these results demonstrate that CGMMV infection activates the JAs signaling pathway and further induces the expression of TFs to initiate downstream gene expression. 

### 2.8. The Function of the JAs Signaling Pathway in CGMMV Infection

To further identify the function of the JAs signaling pathway during CGMMV infection, we used CGMMV-based gene silencing as described previously [45]. *Agrobacterium* containing CGMMV-*gfp*, CGMMV-*BgAOS*, or CGMMV-*Bgpds* was infiltrated into the cotyledons of bottle gourd. At 12 dpi, plants inoculated with CGMMV-*Bgpds* showed obvious photobleaching phenotypes, indicating that *pds* was silenced by CGMMV-induced gene silencing (Figure 8A). Plants inoculated with CGMMV-*gfp* showed mottle and green symptoms, while CGMMV-*BgAOS* showed no obvious disease symptom (Figure 8A). Samples were taken and performed using Western blot and qRT-PCR detection. Compared with CGMMV-*gfp*, CP accumulation was substantially reduced in CGMMV-*BgAOS* (Figure 8B). qRT-PCR also showed that the mRNA level of the *AOS* gene was significantly reduced compared with CGMMV-*gfp*, though not much compared with the mock control (Figure 8C). These results indicated that the downregulation of *AOS* mRNA led to a decrease in CGMMV accumulation, implying that JAs may contribute to CGMMV infection in bottle gourd. 

## 3. Discussion

CGMMV mainly infects cucurbit crops, causing serious damage to the cucurbit industry. Although the symptoms of CGMMV vary between different cucurbit species and cultivars, the classic symptoms are leaf mottling, green, mosaic, and fruit malformation. In a previous study, we found that CGMMV was one of the most prevalent viruses in bottle gourd in Guangdong province of China, causing serious damage to the bottle gourd industry [43]. High-throughput deep sequencing has been extensively used to investigate the interaction between biotic or abiotic stimuli and plants, facilitating the identification of host gene expressions using distinct treatments. Metabolome analysis is another potent methodology for examining changes in metabolites, which are regulated by gene expression and play a crucial role in plant growth, development, differentiation, and defense. Several studies have utilized RNA-seq or whole-genome bisulfite sequencing to identify the genes’ expressions or the DNA methylation level of watermelon leaves or fruit [38,42,46]. However, combined transcriptome and metabolome analysis of bottle gourd with CGMMV infection has not been reported. In this study, transcriptome and metabolome analysis revealed that DEGs and DAMs are mainly involved in pathways related to the biosynthesis of secondary metabolites and plant hormone signal transduction. The JAs signaling transduction pathway was found to be significantly activated, and silencing of the *AOS* gene, the key enzyme in the JAs synthesis pathway, decreased CGMMV accumulation, implying that JAs may play a role in facilitating CGMMV infection. 

In our study, 1929 DEGs were found via transcriptome analysis: 1397 DEGs were upregulated and 532 were downregulated. GO and KEGG pathway enrichment showed that these DEGs are mainly involved in metabolic pathways, the biosynthesis of secondary metabolites, plant hormone signal transduction, and plant–pathogen interaction pathways. A previous study on the impact of CGMMV infection on watermelon leaves and fruit also demonstrated that the DEGs were involved in photosynthesis, plant–pathogen interactions, secondary metabolism, and plant hormone signal transduction [38,42]. These studies uncovered that the targeting of these pathways by CGMMV could potentially serve as the underlying cause of disease symptoms in cucurbit crops. 

We found that CGMMV activates hormone signal transduction pathways, like JAs, ethylene, and brassinosteroid, while the response of the salicylic acid pathway was not obvious. JAs are phytohormones playing an important role in plant growth and development. JA and its derivates are synthesized in chloroplast, peroxisome, and cytoplasm. Transcriptome analysis showed that all of the genes involved in the JAs synthesis pathway were upregulated, and the accumulation of JA and the bioactive JA-Ile hormones was increased, demonstrating that the JAs signaling pathway was activated with CGMMV infection. After being transported to the nucleus, JAs activate the degradation of JAZ protein via the 26S proteasome pathway. A previous report showed that the cucumber mosaic virus (CMV) 2b protein interacts and represses the JAZ protein to manipulate JAs hormone signaling to attract insect vectors [32]. In this study, we found that the expression of JAZ was increased about three times in CGMMV-infected bottle gourd, and further experiments are needed to find out the reason. 

The degradation of JAZ protein releases transcription factors (TFs) and initiates downstream gene expression. TFs involved in the JAs signaling pathway include Mycs, Mybs, NACs, WRKYs, and ERFs. Here, we have identified an increase in the expression of most TFs related to the JAs signaling pathway upon CGMMV infection. The expression of basic helix-loop-helix TF *Myc2* and *Myc3* increased about 2 times and 32 times, responding to CGMMV attack. Reports have shown that both Myc2 and Myc3 are the targets of the JAZ protein and regulate the expression of various subsets of JA-responsive genes [47,48,49]. Furthermore, we showed that some Myb TFs were also induced after CGMMV infection, like *Myb13* (more than 4 times), *Myb62* (more than 30 times), and *Myb86* (10 times). Myb13 has been reported to be a transcriptional activator and enhances the expression of fructosyltransferase to synthesize fructans in wheat [50]. Transgenic overexpression of *Myb48* improved the drought tolerance of *Arabidopsis* plants, indicating that Myb48 may be involved in the drought stress response [51]. *Myb62* regulates the phosphate starvation response by affecting gibberellic acid (GA) biosynthesis [52]. Myb77 interacts with auxin response factors (ARFs) and is involved in the auxin response [53]. In wheat, the expression of *Myb86* was induced by various hormones and cold treatments [54]. *NAC* TF was induced about nine times, but the function of NAC TF is largely unknown. 

WRKY TFs represent another large TF family which consists of 89 members in *Arabidopsis*. Some WRKY TFs are regulated by the JAs signaling pathway, like WRKY22, WRKY50, WRKY57, WRKY70, and WRKY89 [55,56,57,58,59]. A previous study indicated that WRKY3 and WRKY6 are elicited upon herbivore feeding, and silencing *WRKY3* or *WRKY6* decreases the resistance of *Nicotiana attenuate* in herbivores [60]. However, in this study, we found that the expression of *WRKY6* was reduced, demonstrating that WRKY6 may play a varied role in different plants under diverse stresses. Various studies have reported that *WRKY18* was induced upon abiotic or biotic stress [61,62,63,64], and we also confirmed that CGMMV also facilitated *WRKY18* expression. Moreover, we found that the expression of *WRKY31* was upregulated with CGMMV infection. In apple, *WRKY31* was induced after SA treatment, and the ectopic expression of WRKY31 increased the resistance of *Arabidopsis* and *Nicotiana benthamiana* to flg22 and *Pseudomonas syringae tomato* (*Pst DC3000*) [65]. Further, *WRKY68* was induced up to 10 times with CGMMV infection, implying that WRKY68 may play an important role during viral infection. However, the function of WRKY68 under abiotic and biotic stress remains unclear. 

Some ERF TFs are also induced via JAs signaling. We found that the expression levels of *ERF1*, *ERF1b*, *ERF-AP2*, *ERF-2*, and *ERF-2b* were significantly increased, indicating that these TFs may be involved in the antiviral response. ERF1 was identified as a member of the AP2 transcription factor family and was found to function dependently on JAs and/or ET during *Botrytis cinerea* infection [66]. We found that *ERF-RAP2* was induced to nearly 1000 times upon CGMMV infection, indicating the importance of *ERF-RAP2* during viral infection. A previous report has shown that RAP2 increases the resistance of *Arabidopsis* to *Pseudomonas syringae* [67]. *ERF2* was also induced upon CGMMV infection, though not as much as *ERF-RAP2*. Studies showed that *ERF2* was significantly upregulated in tomato plants infected with *Stemphylium lycopersici*, and SA and JA accumulation were lower in *ERF2*-silenced plants [68]. However, the expression of *ERF6* was downregulated under CGMMV stress. On the contrary, a previous report showed that JA/ET-responsive genes were upregulated in *ERF6* transgenic *Arabidopsis* plants, indicating that ERF6 may positively regulate the JAs signaling pathway [69]. Taken together, although a variety of reports have studied the functions of some TFs, the specific function of these TFs during CGMMV needs further research. 

Phytohormones play an important role in the defense response to most biotic and abiotic stimuli, but viruses have evolved varied strategies to avoid host immunity. Although numerous studies have reported the interactions between plants and viruses, the role of phytohormones is still obscure. JAs signaling may play a protective role against plant viruses, like beet curly top virus (BCTV) and CMV [32,35]. In this study, we found that the silencing of *AOS* reduced CGMMV accumulation, implying that JAs signaling may play a negative role in the defense response of bottle gourd to CGMMV. Consistently, JAs also contribute to TMV infection in *N* gene-containing tobacco [36]. Further research may focus on the mechanism behind this, like the interactions between viral proteins and JAs signaling pathway components. 

## 4. Materials and Methods 

### 4.1. Plant Growth Conditions

Bottle gourd (*Lagenaria siceraria* cv. Pugua no. 1) plants were grown in a climate chamber with a 14 h/10 h light/dark photoperiod at 25 °C. 

### 4.2. Virus Inoculation and Detection

The cotyledons of bottle gourd were infiltrated with *Agrobacterium* GV3101 containing pCB301-CGMMV infectious cDNA clone [44]. At 12 dpi, the upper leaves were taken and subjected to Western blot or RT-PCR with CGMMV CP antibody or CGMMV CP specific primer (Appendix A). 

### 4.3. Total RNA Extraction, cDNA Library Construction, and Deep Sequencing

At 12 dpi, approximately 0.1 g of the upper leaves from the mock plants and the CGMMV-infected bottle gourd, with three replicates each, was collected for RNA deep sequencing. Total RNA of the mock plants and CGMMV-infected bottle gourd was extracted using Trizol Reagent (Takara, Dalian, China). DNA contamination was digested using RNase-free rDNase (Transgene, Beijing, China). RNA concentration and integrity were evaluated using a Qubit 2.0 fluorescence spectrometer (Thermo Fisher, Waltham, MA, USA) and Agilent 2100 bioanalyzer (Agilent, Santa Clara, CA, USA). Then, the mRNA of the total RNA was enriched using Oligo dT magnetic beads before the construction of the cDNA library according to the manufacturer’s instructions (Illumina, San Diego, CA, USA). mRNA was fragmented by adding fragmentation buffer; then, the fragmentated mRNAs were used as templates for the synthesis of first-strand cDNA with random hexamers. Double-stranded cDNA was synthesized by adding reaction buffer, dNTPs, and DNA polymerase, followed by purifying with AMPure XP beads (Beckman Coulter, Brea, CA, USA). The purified double-stranded cDNA was end-repaired, and we added an “A” base and sequencing index adapter. The treated double-stranded cDNA was enriched with AMPure XP beads and formed the cDNA library. Deep sequencing was performed using the Illumina HiSeq 2000 platform (Illumina, USA) by Wuhan Metware Biotechnology Co., Ltd. (www.metware.cn, Wuhan, China). 

### 4.4. Transcriptome Analysis

Raw data generated by deep sequencing were treated by removing the read adapter sequences and low-quality reads using Fastp [70]. Clean data were mapped to the bottle gourd genome (http://cucurbitgenomics.org/ftp/genome/BottleGourd/USVL1VR-Ls/) (accessed on 5 June 2022) using hierarchical indexing for spliced alignment of transcripts (HISAT) [71]. Stringtie [72] was used to reconstruct the transcriptome scripts and estimate the expression levels of each mRNA in different samples. Before comparing DEGs between different samples, transcripts were normalized by calculating fragments per kilobase of exon per million mapped reads (FPKM) using RSEM software (v.1.3.1) [73]. DESeq2 [74] was used to analyze the DEGs, with a false discovery rate (FDR) < 0.05 and |log2FC| ≥ 1. GO (http://www.geneontology.org/) (accessed on 5 June 2022) and KEGG (https://www.genome.jp/kegg) (accessed on 5 June 2022) pathway enrichment was used to analysis the DEGs. 

### 4.5. RNA-Seq Validation Using qRT-PCR

To validate the transcriptome result, the total RNA of the mock and CGMMV-treated bottle gourd was extracted and the genomic DNA contamination was digested using RNase-free rDNase before reverse transcription using the PrimeScript™ RT reagent Kit with a gDNA Eraser (Takara, Dalian, China). Twenty DEGs were selected for qRT-PCR detection after the analysis of the transcriptome results. The primers used for real-time PCR of the 20 DEGs, the genes involved in JAs synthesis, and the TFs are listed in Appendix A, and the *Histone H3* gene of *Lagenaria siceraria* (*LsH3*) [75] was used as the reference gene. Real-time PCR was performed with TB green Premix Ex Taq (Takara, Japan) using the CFX96 Real-Time System (Bio-Rad, Hercules, CA, USA). 

### 4.6. Metabolome Analysis

Sample preparation, extraction, and metabolite quantification were performed according to the protocol published previously [76] by Wuhan Metware Biotechnology Co., Ltd. Metabolite profiling was conducted using ultra-performance liquid chromatography (UPLC, Shim-pack UFLC Shimadzu CBM30A) and tandem mass spectrometry (MS) (Applied Biosystems 4500 QTRAP). Qualitative analysis of metabolites was performed based on the metware database, and quantitative analysis of metabolites was performed via multiple reaction monitoring (MRM) and triple quadrupole mass spectrometry [77,78]. Analyst 1.6.3 was used to analyze the MS data and |log_2_Fold Change| ≥ 1 were filtered as significant DAMs. The heatmap was drawn using R software (v.4.1.0). 

### 4.7. CGMMV-Based Gene Silencing 

The CGMMV infectious cDNA clone was constructed previously [44]. To construct the CGMMV-based gene silencing system, plasmid containing CGMMV infectious cDNA clone was modified, as reported previously [45]. Briefly, the full length of CGMMV was inserted into the pCB301 vector to make a pCB301-CGMMV infectious clone. Then, the sequence of CP promoter was repeated to start the transcription of downstream gene expression in the intergenic region of CP and MP ORF. The phytoene desaturase (PDS) is an essential enzyme during the plant carotenoid biosynthetic pathway, and silencing of PDS results in photobleaching phenotypes in *Nicotiana benthamiana* and cucurbit crops. Fragments of *AOS* and *PDS* of 300 bp in length were amplified from bottle gourd cDNA and inserted into the *BamHI*-digested CGMMV-infectious cDNA clone to make CGMMV-*BgAOS* and CGMMV-*Bgpds*. CGMMV-*gfp* was used as a positive control.

## Figures and Tables

**Figure 1 ijms-24-16566-f001:**
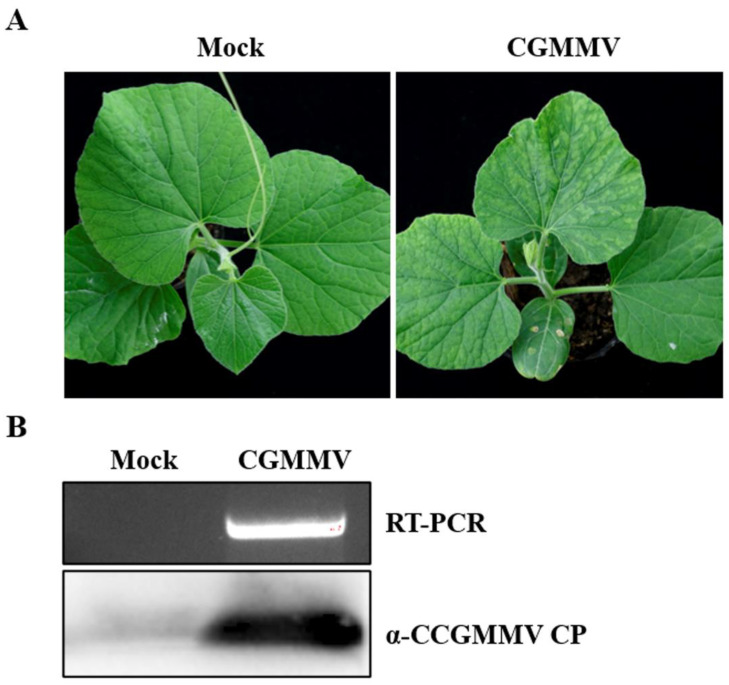
Symptoms of CGMMV in bottle gourd and RT-PCR and Western blot detection of CGMMV. (**A**) Symptoms of CGMMV in bottle gourd. *Agrobacterium* strains containing CGMMV infectious cDNA clones or an empty vector were infiltrated into the cotyledons of bottle gourd. Photos were taken at 12 dpi. (**B**) At 12 dpi, the upper leaves of bottle gourd were taken and detected using CGMMV CP specific primer and CP antibody.

**Figure 2 ijms-24-16566-f002:**
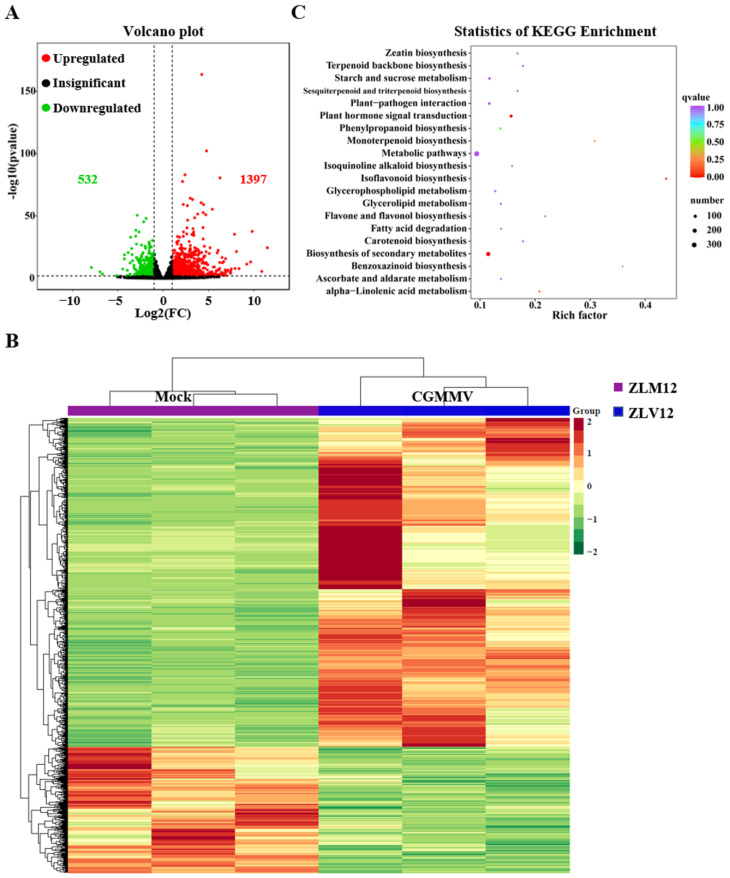
Transcriptome analysis of bottle gourd in response to CGMMV infection. (**A**) Volcano plot showing the DEGs revealed via RNA-seq analysis. Red, green, and black dots represent the upregulated, downregulated, and insignificant DEGs, respectively. (**B**) Clustering heatmap of the DEGs between ZLM12 and ZLV12. The abscissa of each sample represents three replicates. ZLM12 represents mock treatment and ZLV12 represents CGMMV treatment. (**C**) KEGG enrichment analysis of the DEGs.

**Figure 3 ijms-24-16566-f003:**
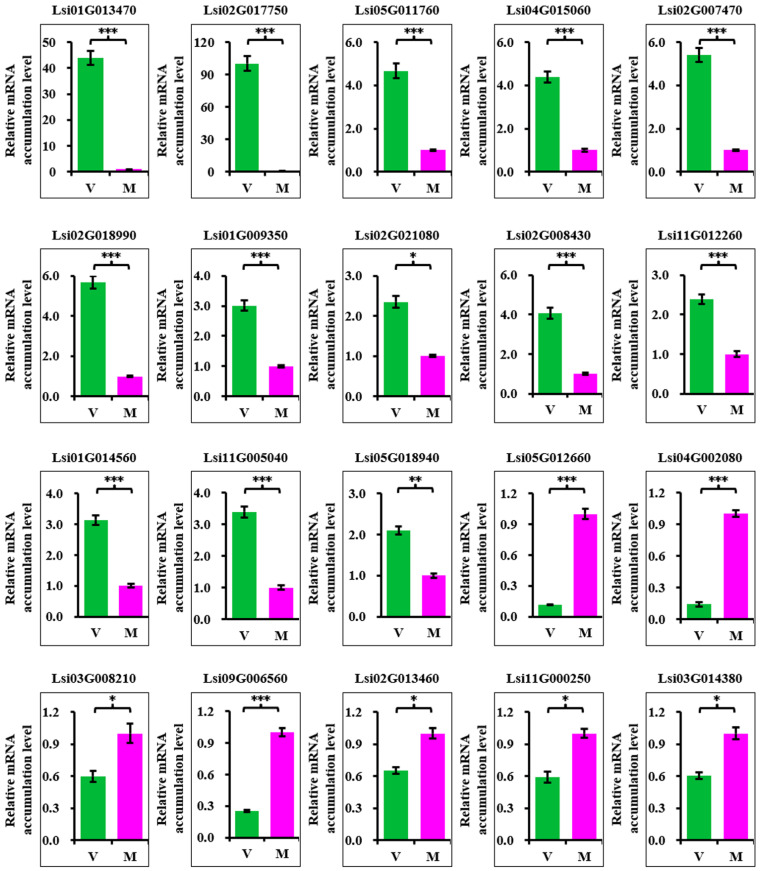
qRT-PCR validation of the selected DEGs revealed via transcriptome analysis. Validation of 20 DEGs using qRT-PCR. The first 13 graphs show the upregulated DEGs, and the last 7 graphs show the downregulated DEGs. V, CGMMV-infected samples. M, mock samples. Annotation of the DEGs is shown in Table 2. Asterisks indicate the *p*-value between mock and CGMMV plants using Student’s *t*-test method. * *p* < 0.05, ** *p* < 0.01, *** *p* < 0.001.

**Figure 4 ijms-24-16566-f004:**
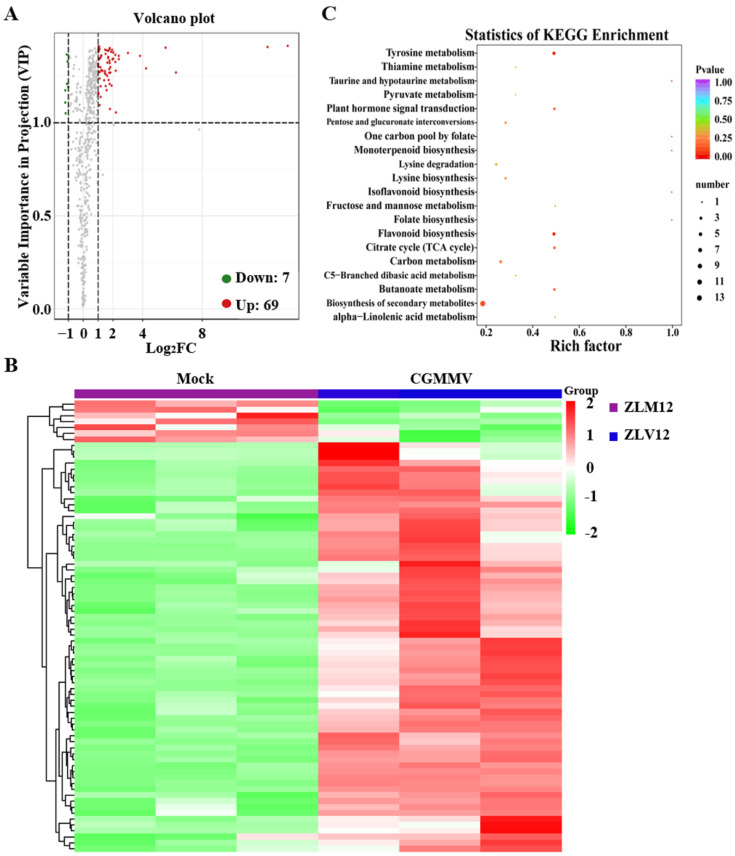
Metabolome analysis of bottle gourd in response to CGMMV infection. (**A**) Volcano plot showing the DAMs between the mock and CGMMV-infected samples. Red, green, and gray dots represent the upregulated, downregulated, and insignificant DAMs, respectively. (**B**) Clustering heatmap of the DEGs between ZLM12 and ZLV12. Each treatment contains three replicates. (**C**) KEGG enrichment analysis of the DAMs.

**Figure 5 ijms-24-16566-f005:**
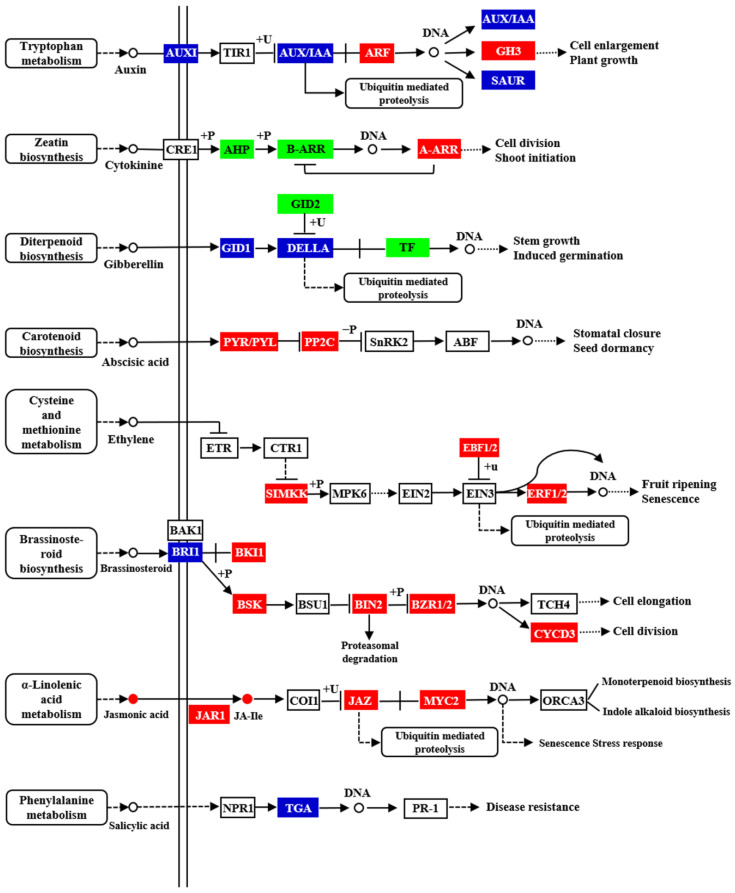
Analysis of hormone signaling transduction in response to CGMMV infection. Joint analyses of hormone signaling pathway in response to CGMMV infection. Red blocks represent upregulated genes, green blocks represent downregulated genes, and blue blocks represent the genes which are not consistent between different replicates of each treatment. Red dot represents that the hormone is up-accumulated.

**Figure 6 ijms-24-16566-f006:**
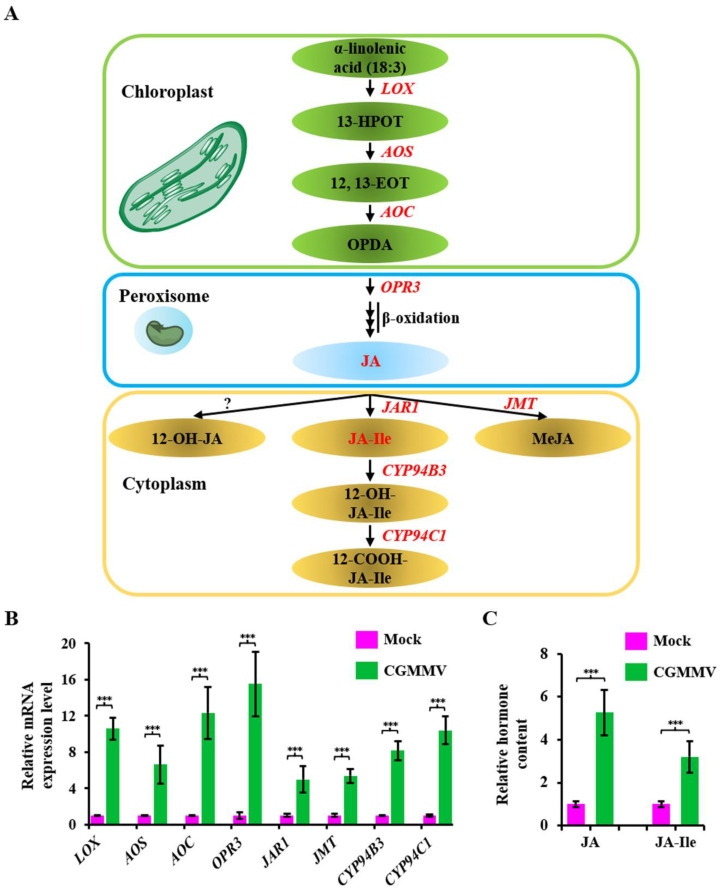
Analysis of DEGs and DAMs involved in JAs synthesis pathway with CGMMV infection. (**A**) Scheme of the JAs synthesis pathway. DEGs and DAMs identified via transcriptome and metabolome analyses are marked in red. (**B**) qRT-PCR verification of the DEGs involved in the JAs synthesis pathway. Asterisks indicate the significance of difference between mock and CGMMV plants using Student’s *t*-test method. *** *p* < 0.001. (**C**) Comparison of the accumulation of JA and JA-Ile in mock and CGMMV-treated bottle gourd. *** *p* < 0.001.

**Figure 7 ijms-24-16566-f007:**
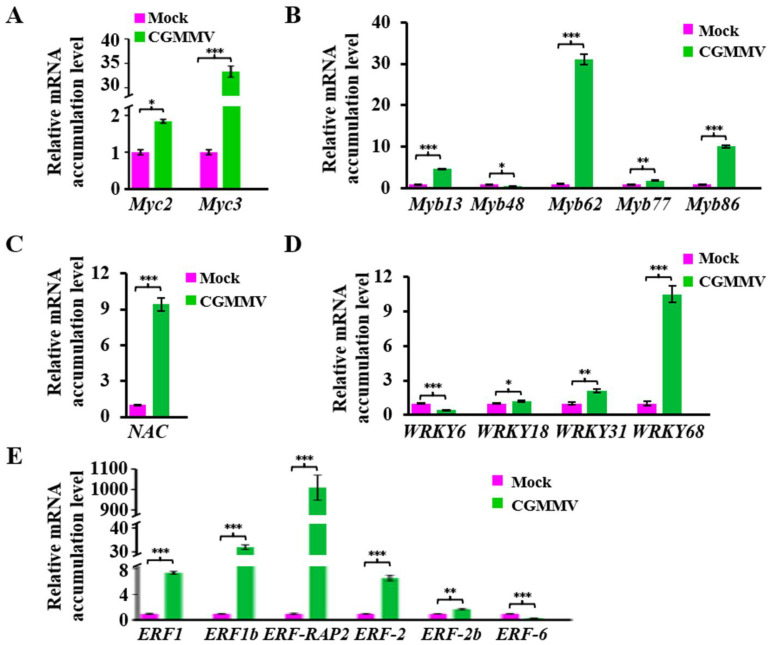
Verification of the expressions of TFs using qRT-PCR. (**A**) Expressions of Myc TFs. *Myc2*, Lsi09G016530. *Myc3*, Lsi07G011090. (**B**) Expressions of Myb TFs. *Myb13*, Lsi05G010160. *Myb48*, Lsi10G013470. *Myb62*, Lsi02G001830. *Myb77*, Lsi04G015930. *Myb86*, Lsi06G015850. (**C**) Expressions of TF *NAC* (Lsi08G004490). (**D**) Expressions of WRKY TFs. *WRKY6*, Lsi05G021100. *WRKY18*, Lsi02G012150. *WRKY31*, Lsi05G014110. *WRKY68*, Lsi06G014830. (**E**) Expressions of ERF TFs. *ERF1*, Lsi06G008160. *ERF1b*, Lsi09G012390. *ERF-RAP2*, Lsi08G002460. *ERF-2*, Lsi07G001170. *ERF-2b*, Lsi03G010920. *ERF-6*, Lsi04G020490. The significance of difference between mock and CGMMV plants was tested via *p*-value using Student’s *t*-test method. * *p* < 0.05, ** *p* < 0.01, *** *p* < 0.001.

**Figure 8 ijms-24-16566-f008:**
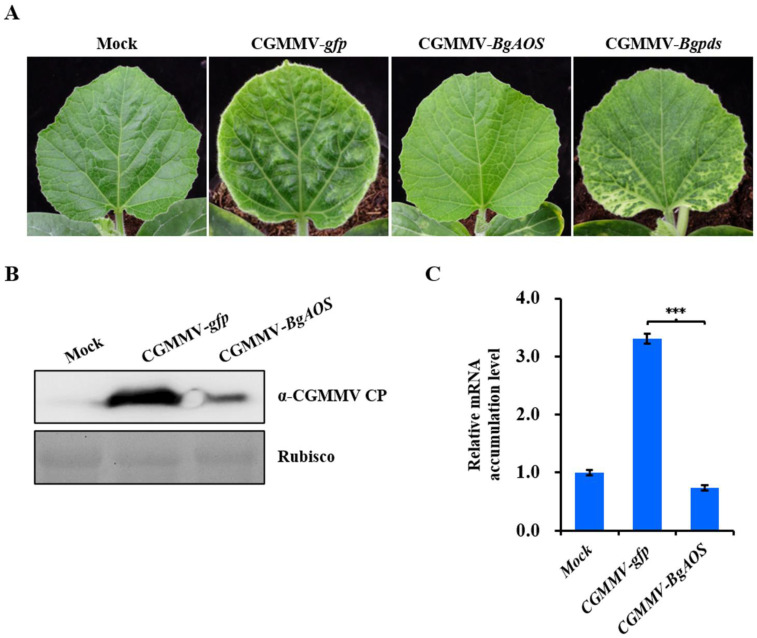
Effect of *AOS* gene silencing on CGMMV infection. (**A**) *Agrobacterium* strains containing empty vector (mock), CGMMV-*gfp*, CGMMV-*BgAOS*, or CGMMV-*Bgpds* were infiltrated into the cotyledons of bottle gourd. Photos were taken at 12 dpi. (**B**) Western blot detection of CGMMV accumulation with CP antibody. Rubisco was used as the equal loading control. (**C**) qRT-PCR analysis of *AOS* mRNA accumulation. The significance of difference between CGMMV-*gfp* and CGMMV-*BgAOS* was tested via *p*-value using Student’s *t*-test method. *** *p* < 0.001.

**Table 1 ijms-24-16566-t001:** Overview of the RNA-seq data of bottle gourd.

Sample	Raw Reads	Clean Reads	Clean Base	Unique Mapped Reads	Q20 (%)	Q30 (%)	GC Content (%)
ZLM121	60,300,774	59,025,240	8.85 G	55,742,483 (94.44%)	97.48	93.02	44.80
ZLM122	50,095,732	49,062,610	7.36 G	46,436,613 (94.65%)	97.43	92.87	45.17
ZLM123	48,349,016	47,355,442	7.10 G	45,101,108 (95.24%)	97.61	93.26	44.86
ZLV121	44,431,754	43,343,210	6.50 G	41,374,202 (95.46%)	97.77	93.61	44.56
ZLV122	51,042,946	50,073,404	7.51 G	46,079,504 (92.02%)	97.50	93.02	44.48
ZLV123	43,970,228	42,963,264	6.44 G	40,631,748 (94.57%)	97.44	92.92	44.92

**Table 2 ijms-24-16566-t002:** Annotation of the selected DEGs for qRT-PCR verification.

Gene ID	Description
Lsi01G013470	Aquaporin PIP2-2-like
Lsi02G017750	Probable calcium-binding protein CML44
Lsi05G011760	Probable membrane-associated kinase regulator 6
Lsi04G015060	Cytochrome P450 81D1-like isoform X1
Lsi02G007470	E3 ubiquitin-protein ligase RMA1H1-like
Lsi02G018990	Heterodimeric geranylgeranyl pyrophosphate synthase small subunit, chloroplastic-like
Lsi01G009350	Probable indole-3-acetic acid-amido synthetase GH3.3
Lsi02G021080	Pectinesterase
Lsi02G008430	Serine/threonine-protein kinase STY8
Lsi11G012260	ACT-like tyrosine kinase family protein
Lsi01G014560	NADPH-dependent aldo-keto reductase, chloroplastic-like
Lsi11G005040	Disease resistance protein (TIR-NBS-LRR class)
Lsi05G018940	Invertase/pectin methylesterase inhibitor family protein
Lsi05G012660	Dehydration responsive element-binding protein 1
Lsi04G002080	MATE efflux family protein
Lsi03G008210	UDP-glycosyltransferase 74F1
Lsi09G006560	DUF4228 domain protein
Lsi02G013460	Scarecrow transcription factor family protein
Lsi11G000250	Protein brassinosteroid insensitive 1
Lsi03G014380	Seed maturation-like protein

## Data Availability

Data is contained within the article and Appendix A.

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
