# Peer review of "Transcriptome and Metabolome Analyses Reveal That Jasmonic Acids May Facilitate the Infection of Cucumber Green Mottle Mosaic Virus in Bottle Gourd"

_ijms, 2023, doi:10.3390/ijms242316566_

Round 1
Reviewer 1 Report
Comments and Suggestions for Authors
The work was written carelessly.
The paper needs to be corrected and resubmitted.
Details of the manuscript are provided below.
Introduction
Abbreviations without the full name are present in the manuscript; even the abstract (Lane 9) begins with an unexplained abbreviation of the name of the virus. The whole manuscript should be checked. Another example is in lane 25, where the name of the AOS gene is not explained, SCF – lane 69 and others throughout the manuscript.
What does CMMV Bgpds mean? CGMMV CP?
What does the LsH3 gene encode?
Lanes 37 and 43
Did the authors not mean for the first time when they wrote "firstly"?
Lane 40 - I think it should be international and not intranational.
The introduction needs updating.
The authors state in lines 74-75 that there is no information in the literature about the role of JA during stress caused by the CGGM virus; however, such data were included in the work Slavokhotova et al. 2021, which analysed the role of various hormones (including JA) in response to stress caused by CGGM virus.
Slavokhotova, A.; Korostyleva, T.; Shelenkov, A.; Pukhalskiy, V.; Korottseva, I.; Slezina, M.; Istomina, E.; Odintsova, T. Transcriptomic Analysis of Genes Involved in Plant Defense Response to the Cucumber Green Mottle Mosaic Virus Infection. Life 2021, 11, 1064. https://doi.org/10.3390/life11101064
Results
The figures with pictures of western blots and qPCR results are not prepared properly. The original pictures in the supplementary materials Figure 1B are not described. Moreover, they are cropped. These data should be prepared according to the guidelines https://www.mdpi.com/journal/cells/instructions#oriimages.
Figures 2 and 4. Panels A and C are too small and illegible. They should be increased at the expense of panel B, which will still be readable when reduced.
Panel B in the legend describing colours - Instead of or in addition to the description of the sample numbers ZLM12 and ZLV12, Mock and Virus should be added to make it easier to read.
Figure 2 – in the description of this figure, there is a mistake in the plant name – there is only bottle, and the gourd is lacking.
The description of Figures 3 and 7 lacks information on what the green and red colours stand for.
Figure 8 - Why are the results for the CMMV Bgpds not posted?
Lanes 134-135
This sentence is rather a description of methods; it does not describe the results.
Lane 162-166. These are not complete sentences. The verb is missing.
The sentence in lines 168-171 is rather for a discussion
The whole of paragraph 2.4 does not describe the results.
Whether the authors expect readers to interpret Figure 3?
There is no information on whether and what differences were observed between control and infected plants. Were these differences statistically significant?
The statement in line 170 " the degree of variation differed among different DEGs" is insufficient.
Lanes 199-201 –contain only an enumeration of plant hormones and are therefore superfluous. It is sufficient to state for which expression changes and to observe changes in their amount after virus infection.
Lane 231
- …induced about two times .... Induced should be in lowercase as it is a continuation of the sentence for page 10.
Figure 7 - add descriptions to gene names or refer to relevant tables.
Discussion
Lines 296 to 298 are an exact copy of a sentence from the results - lines 144-146
Lanes 298-303 - Sentences need to be rewritten.
Lane 308 - nearly all – it should be precise
Lane 315 - results not presented should not be invoked. Either remove this sentence or show the results. This claim cannot be addressed.
lines 338-342
The other reason why this difference was observed might be because a different plant species was tested, not tobacco.
Lane 338 -
Melanie is the author's name, not her surname. When quoting the family name, the sentence should be rewritten.
Materials and methods
Paragraph 4.5 qRT-PCR
The authors provide the number of genes for which qPCR was performed. It is not enough to refer to table S1. It could be mentioned that qPCR was performed for 20 DEGs selected after the analysis of transcriptomes, ... DEGs involved in the synthesis of JA and for ... TFs listed in table S1.
Paragraph 4.7.
The method should be briefly described, especially since the work to which readers are referred is in Chinese.
Author Response
Please see the attachment, thank you.

Reviewer 2 Report
Comments and Suggestions for Authors
In my opinion for the manuscript to be accepted, some points must be improved.
Major points:
1- The results of the expression of TFs related to the JAs signaling pathway under CGMMV stress are not convincing. Statistical analysis (p-values) is necessary to assess whether the increase or reduction in gene expression is or not significantly relevant, especially for those FTs that showed few increases under the infection context. Add this statistical analysis to all figures that have graphs.
2- The M&M section does not provide all the necessary details of the experimental design. For example, the number of samples and the replicates used for transcriptomic and metabolomic analyses were not described at any time in the M&M section. How many upper leaves per plant and replicates were evaluated? Part of this information is only commented on in the results section. Furthermore, the authors must describe the methods in more detail. Line 439 “fragments were amplified: What enzyme was used for it? What method (golden gate, assembly) was used for the fragment insertion into the infectious cDNA clone? Details about restrict enzymes used were not described. What does mean pds? A reader who is not familiar with these terms will not be able to follow the study. This information must be presented: Infection with the CGMMV vector harboring PDS (the phytoene desaturase) sequences of 69–300 bp in length in the form of sense-oriented or hairpin cDNAs resulted in photobleaching phenotypes in N. benthamiana and cucurbits by PDS silencing. This is just one example, but several others are found repeatedly throughout the manuscript.
Minor points:
1- The first time the acronyms are presented, their meaning must be accompanied. “JAs and AOS” in the abstract are some examples.
2- Line 66, delete the word “protein” after CP.
3- What is the economic importance of the bottle gourd for China? Why was this plant selected for the study? How much is the loss in global and Chinese production by CGMMV infection in this cultivar? All these points must be presented in the introduction section.
4- Fig. 3, highlights which of the graphs are related to upregulation and downregulation.
5- Lines 201-203, “showed that the jasmonic acid and brassinosteroid signal transduction pathways were remarkably induced, while changes of the other hormone signaling pathways were not obvious”. It's not obvious to me, and the reader can't come to that same conclusion just by looking at figure 5.
6- Line 257, why wasn't this comparison also made with CGMMV-Bgpds?
7- Figure 8C, this result is not convincing. Apparently, no significative differences between AOS mRNA accumulation compared with empty vector were observed.
8- Fig. 8B, rubisco image must be improved. What was this stained with?
9- Fig. 7, indicates what each color of the bars means.
Round 2
Reviewer 1 Report
Comments and Suggestions for Authors
The authors have made the expected corrections and responded to the comments.
I have no further comments.
Author Response
Dear Reviewer:
Thanks a lot for reviewing our manuscript.
Reviewer 2 Report
Comments and Suggestions for Authors
Many points have been improved by the authors. however, some questions were only answered and not added to the article, for example: "Response: Thanks. In JA and BR signal pathway, the number of DEGs and DAMs are relatively more than other pathways." Please make this clear in the text for the readers. The same applies to: Response: Thanks. The first 13 graphs show the upregulated genes, and the last 7 graphs show the downregulated genes.
Author Response
Dear Reviewer:
Thanks a lot for reviewing our manuscript, and we also appreciate your suggestible review of our manuscript. We have responded to your comments below. The revisions are highlighted in red in manuscript.
Comments: Many points have been improved by the authors. however, some questions were only answered and not added to the article, for example: "Response: Thanks. In JA and BR signal pathway, the number of DEGs and DAMs are relatively more than other pathways." Please make this clear in the text for the readers. The same applies to: Response: Thanks. The first 13 graphs show the upregulated genes, and the last 7 graphs show the downregulated genes.
Response: Thanks for your favorable view of our manuscript. We have updated the above two responses, please see result 2.6 and Figure 3 legend, respectively.